# Ultrafast electronic response of graphene to a strong and localized electric field

Elisabeth Gruber[1], Richard A. Wilhelm[1,2], Rémi Pétuya[3], Valerie Smejkal[1], Roland Kozubek[4], Anke Hierzenberger[4], Bernhard C. Bayer[5], Iñigo Aldazabal[6], Andrey K. Kazansky[3,7], Florian Libisch[8], Arkady V. Krasheninnikov[2], Marika Schleberger[4], Stefan Facsko[2], Andrei G. Borisov[9], Andrés Arnau[3,6,10] & Friedrich Aumayr[1]

The way conduction electrons respond to ultrafast external perturbations in low dimensional materials is at the core of the design of future devices for (opto)electronics, photodetection and spintronics. Highly charged ions provide a tool for probing the electronic response of solids to extremely strong electric fields localized down to nanometre-sized areas. With ion transmission times in the order of femtoseconds, we can directly probe the local electronic dynamics of an ultrathin foil on this timescale. Here we report on the ability of freestanding single layer graphene to provide tens of electrons for charge neutralization of a slow highly charged ion within a few femtoseconds. With values higher than $10^{12}\,A\,cm^{-2}$, the resulting local current density in graphene exceeds previously measured breakdown currents by three orders of magnitude. Surprisingly, the passing ion does not tear nanometre-sized holes into the single layer graphene. We use time-dependent density functional theory to gain insight into the multielectron dynamics.

[1] TU Wien, Institute of Applied Physics, 1040 Vienna, Austria. [2] Helmholtz-Zentrum Dresden-Rossendorf (HZDR), Institute of Ion Beam Physics and Materials Research, 01328 Dresden, Germany. [3] Donostia International Physics Centre (DIPC), 20018 Donostia-San Sebastian, Spain. [4] Universität Duisburg-Essen, Fakultät für Physik and Cenide, 47048 Duisburg, Germany. [5] University of Vienna, Faculty of Physics, Boltzmanngasse 5, 1090 Vienna, Austria. [6] Centro de Fisica de Materiales (CFM), Centro Mixto CSIC-UPV/EHU - MPC, 20018 Donostia-San Sebastian, Spain. [7] IKERBASQUE, Basque Foundation for Science, 48013 Bilbao, Spain. [8] TU Wien, Institute for Theoretical Physics, 1040 Vienna, Austria. [9] CNRS-Université Paris Sud, Institut des Sciences Moléculaires d'Orsay - UMR 8214, 91405 Orsay Cedex, France. [10] Departamento de Fisica de Materiales UPV/EHU, Facultad de Quimica, 20018 Donostia-San Sebastian, Spain. Correspondence and requests for materials should be addressed to E.G. (email: egruber@iap.tuwien.ac.at) or to F.A. (email: aumayr@iap.tuwien.ac.at) or to A.G.B (email: andrei.borissov@u-psud.fr) or to A.A. (email: andres.arnau@ehu.es) or to M.S. (email: marika.schleberger@uni-due.de).

ts exceptionally high carrier mobility[1] makes graphene a promising material for future electronic applications. The linear Dirac-like dispersion and the associated constant high carrier velocity promise the realization of ultrafast devices in electronics[2], optics[3] or even q-bits based on nitrogen vacancies[4]. The ultra-short timescales involved, $<1$ ps, provide stringent requirements on material properties. Direct current measurements on supported single layer graphene (SLG) reveal breakdown currents due to Joule heating larger than in copper, with densities of about $10^8$–$10^9$ A cm$^{-2}$ (refs 5–8). Note that these measurements include substrate and finite size effects, which may increase the breakdown current as compared with pristine freestanding graphene. Indeed, heat dissipation via the substrate seems essential to achieve these numbers[9,10]. Photoexcitation measurements reveal efficient excited carrier relaxation within a few hundred femtoseconds[11,12] yet probe the lattice on the length scale of the optical wavelength. Moreover, large electric fields[13,14] and collisions with energetic particles[15–28], which allow for tuning the properties of graphene, further modify the response of the material. A reliable tool to locally probe the timescales of the electronic response of pristine graphene to large fields is thus urgently needed.

One way to measure the short-time response of materials is the irradiation with highly charged ions (HCIs), which results in an extremely large, local external field: the Coulomb field of an approaching HCI. A charge state of $q_{in} = 35$ implies a local electric field strength of $1.8 \times 10^{11}$ V m$^{-1}$ at 5 Å distance from it. Achieving the same local field strength using laser fields would require power densities above $10^{17}$ W cm$^{-2}$, a field strength where non-destructive measurements become challenging. Previous work on scattering of HCI from solid surfaces or their transmission through freestanding carbon membranes reported unexpectedly large charge capture within 5–30 fs (refs 29–34).

In this contribution, we take the final step and present the results for the ultimately thin carbon target, a freestanding single layer of graphene. Measurements of the charge state and energy of the transmitted ions and time-dependent density functional theory (TDDFT) calculations show that large number of electrons are extracted from a very small surface area, which implies a high local surface current density. We identify a multielectron process and estimate the relevant timescales for charge transfer along the graphene layer. For example, while passing through SLG, the HCI with initial charge state $q_{in} = 30$ captures $\sim 25$ electrons on a timescale of a few fs. Based on the experimental data, a lower bound for the current densities reached locally within a nm$^2$ area is at least $10^{12}$ A cm$^{-2}$ exceeding the breakdown current densities reported so far[5–8] by three orders of magnitude.

## Results

**Exit charge state analysis.** The amount of charge transferred to the HCI can be estimated by measuring the distribution of exit charge states $q_{out}$ and the energy of highly charged Xe ions after transmission through SLG (for details of sample preparation, see Supplementary Note 1). Initial ion charge states of $10 \leq q_{in} \leq 35$ and velocities below 0.5 nm fs$^{-1}$ were used. Ions are transmitted through SLG under normal incidence and analysed with respect to their charge state and kinetic energy by an electrostatic analyser (Fig. 1c). Typical transmission spectra (Fig. 1a) show a distribution of different exit charge states, with a mean value $\bar{q}_{out}$ shifted towards smaller exit charge states for slower ions. To extract the abundances and widths of every single peak, the spectra have to be deconvoluted with the analyser function, since the spectra are broadened by the design of the electrostatic analyser (details on the data evaluation can be found in ref. 35). The corrected abundances follow a symmetric Gaussian function with

a full width at half maximum of three to five electrons as a result of final de-excitation processes. From these Gaussian fits the mean value $\bar{q}_{out}$ is extracted. Even for the smallest velocities (largest interaction times) used in our experiment ($v_{min} = 0.13$ nm fs$^{-1}$), $\bar{q}_{out}$ remains considerably larger than the equilibrium charge state of a Xe ion in a solid target ($q_{eq} \sim 1$)[36], indicating incomplete neutralization.

Depending on the initial charge state the HCI captures and stabilizes between 20 and 30 electrons during its transmission through graphene (Fig. 1e). The experimentally observed average electron capture is extracted from the mean exit charge state of each transmission spectrum. The transmission time through graphene can be defined by $\tau = d_{eff}/v$, where $v$ is the ion velocity. The effective interaction length $d_{eff}$ corresponds to the projectile-surface distance where the electron transfer processes between the HCI and graphene take place. The values of $d_{eff}$ can be obtained from the TDDFT calculations which yield for $q_{in} = 20$ a value of $d_{eff} \simeq 9$ Å. This is in good agreement with predictions of the classical over the barrier model[37].

The number of captured and stabilized electrons (they are not reemitted due to autoionization processes) as function of interaction time is shown in Fig. 1e. The data can be well fitted by the simple expression

$$q_{in} - \bar{q}_{out} = q_{in}\left(1 - e^{-\tau/\tau_n^{exp}}\right) \qquad (1)$$

with an effective neutralization time constant $\tau_n^{exp}$. Using $d_{eff} = 9$ Å we obtain $\tau_n^{exp} = 2.1$ fs for $q_{in} = 35$, and $\tau_n^{exp} = 1.4$ fs for $q_{in} = 20$, respectively. The performed TDDFT study (for details, see Supplementary Note 2 and Supplementary Fig. 4) shows a multielectron character of the charge transfer that can explain the experimentally observed strong reduction of the charge state of the HCI. We calculate that HCIs with initial charge states $q_{in} = 10$, 20 and 40 capture $\sim 9$, $\sim 17$ and $\sim 34$ electrons during the passage through the graphene layer, respectively.

Since at least $q_{in} - \bar{q}_{out}$ electrons are transferred from the graphene sheet to the HCI during the interaction, the lower bound for the electron transfer rate is given by $(q_{in} - \bar{q}_{out})/\tau$, which corresponds to $10^{16}$ electrons per second as can be estimated from the experimental data. This corresponds to a local current of $I \gtrsim 1.5$ mA. The TDDFT study shows on the one hand that the charge is extracted from the graphene area with lateral radius $R = 5$ Å around the impact point that we can define as an interaction region (Figs 1b and 2), and on the other hand that the electron flow along the graphene layer compensates the electron extraction by the HCI on the timescale of the collision (fs), otherwise the neutralization of the projectile would be stopped by the local charging of the target. The latter is in accord with experimental data that shows the absence of the post-collisional defects that would result from Coulomb explosion (see the section Transmission electron microscopy results). The electrons moving along graphene enter the interaction region crossing the surface $S = 2\pi R h$ ($h = 3.4$ Å is the width of the graphene layer that we estimate from the interlayer spacing of graphite[38]). Thus, a time averaged electron current density $J = I/S$ in the graphene plane exceeding $\sim 6 \times 10^{11}$ A cm$^{-2}$ is reached. This value, however, is just a lower limit because it only accounts for the electrons captured by the projectile. Indeed, because of the interaction with HCI $N_{vac}$ electrons will also be emitted into vacuum. We calculate that for each captured electron approximately one additional electron is emitted. $N_{vac}$ can be even larger if one accounts for Auger processes involving tightly bound electronic shells of the projectile. Estimations as high as $N_{vac} \approx 3 \times q_{in}$ (ref. 39) have been reported.

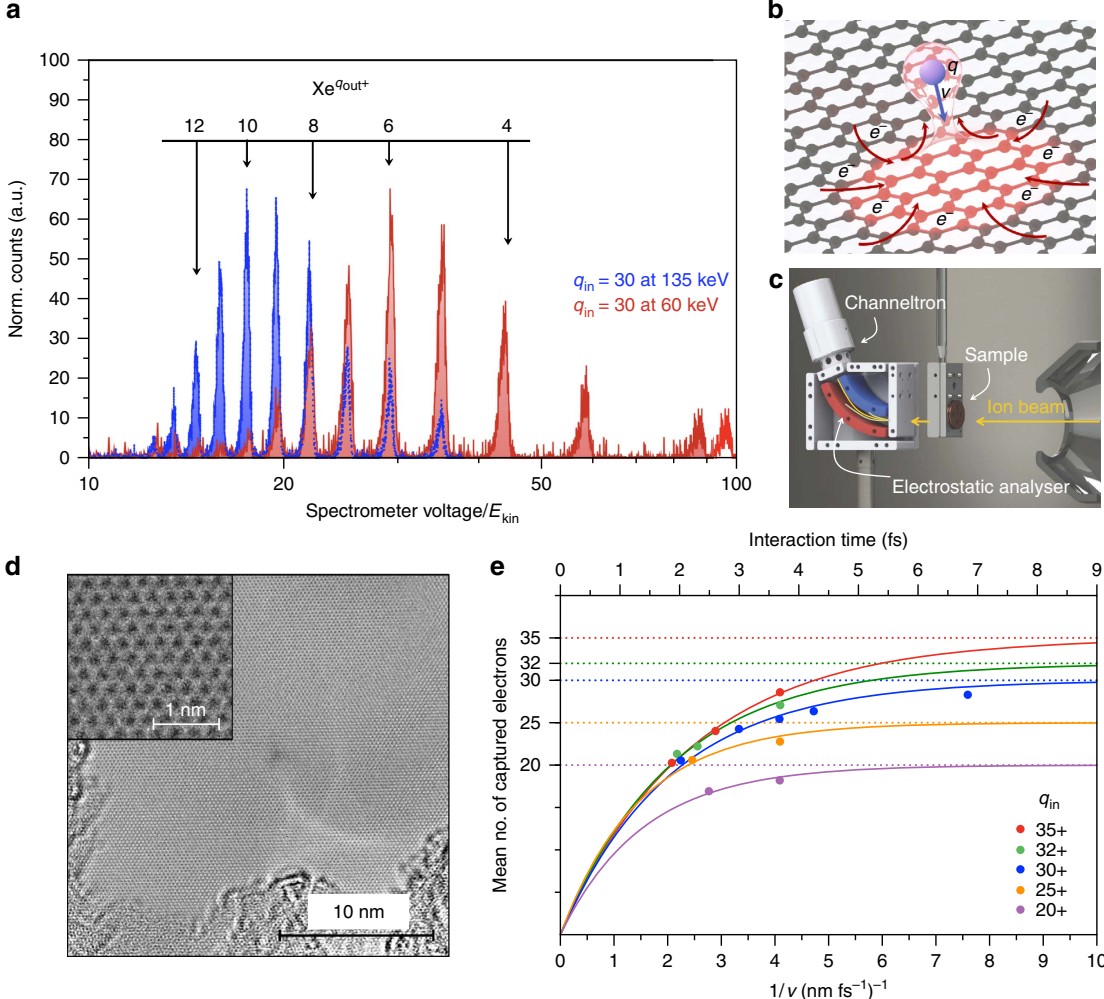

**Figure 1 | Experimental scheme and results.** (**a**) Measured spectra of a $Xe^{30+}$ beam at kinetic energies of 135 and 60 keV (blue and red, respectively) transmitted through a freestanding SLG sheet. Exit charge states $q_{out}$ are calculated from the spectrometer voltage of the electrostatic analyser. The exit charge state distribution shifts towards smaller average exit charge $\bar{q}_{out}$ for slower ions. (**b**) Schematic of the interaction process between freestanding SLG and an approaching highly charged ion (HCI). The HCI extracts a lot of charge from a very limited area on the femtosecond time scale leading to a temporary charge-up of the impact region. (**c**) Sketch of the experimental set-up with the target holder and electrostatic analyser. (**d**) TEM image of a freestanding monolayer of graphene after irradiation with $Xe^{40+}$ ions at 180 keV with an applied fluence of $10^{12}$ ions per $cm^2$ (about six impacts on the shown scale). No holes or nanosized topographic defects could be observed. The inset shows the intact hexagonal structure of graphene. (**e**) Average number of captured and stabilized electrons ($q_{in} - \bar{q}_{out}$) after transmission of $Xe^{q_{in}+}$ ions through a single layer of graphene as a function of the inverse projectile velocity for different incident charge states. Fits to the experimental data points assume a continuous neutralisation following an exponential function. Neutralisation time constants of a few femtoseconds can be extracted.

**Transmission electron microscopy results.** Failure to sufficiently resupply the lost charge and to dissipate the absorbed energy on a timescale small compared with lattice vibrations would result in Coulomb explosion tearing large holes (of the order of 10 nm) into the SLG, as we have observed for carbon nano-membranes[35,40]. Despite the possible self-healing of localized defects in graphene[41], such extended structural modifications should be detectable using transmission electron microscopy (TEM) or scanning transmission electron microscopy (STEM), yet careful investigation of the irradiated SLG does not reveal any nanometre-sized defect structures. Note that for freestanding SLG used in our study, in contrast to supported graphene layers, the defect formation because of a collision cascade in the substrate[42] is not operative. In our case, elastic collisions (nuclear stopping) may cause direct knockout of carbon atoms, but less than one carbon atom is sputtered on average by a 10–100 keV Xe ions according to Lehtinen *et al.*[43]. Even if point defects are produced, they will likely disappear due to dissociation of ubiquitous

hydrocarbon molecules[41]. In Fig. 1d, a typical TEM image of a freestanding monolayer of graphene after irradiation with $Xe^{40+}$ ions with a kinetic energy of 180 keV is shown. The applied fluence of $10^{12}$ ions per $cm^2$ corresponds to approximately six ion impacts within this 25 nm × 25 nm frame. No rupture could be detected. This is in strong contrast to ultrathin polymeric carbon nanomembranes, where, because of low electron mobility, creation of pores with diameters of up to a few nanometre was observed after exposure to HCIs[35,40]. The absence of any traces of large-scale lattice deformations thus confirms the intrinsic ability of suspended SLG to locally sustain exceptionally high current densities, even though it cannot efficiently diffuse heat to a substrate[9,10].

**Local current density.** To illustrate the way the electronic processes take place, in Fig. 2, we show snapshots of the current density for a $q_{in} = 20$ projectile incident at graphene with velocity $v = 0.87$ nm fs$^{-1}$. Results of the TDDFT calculations are

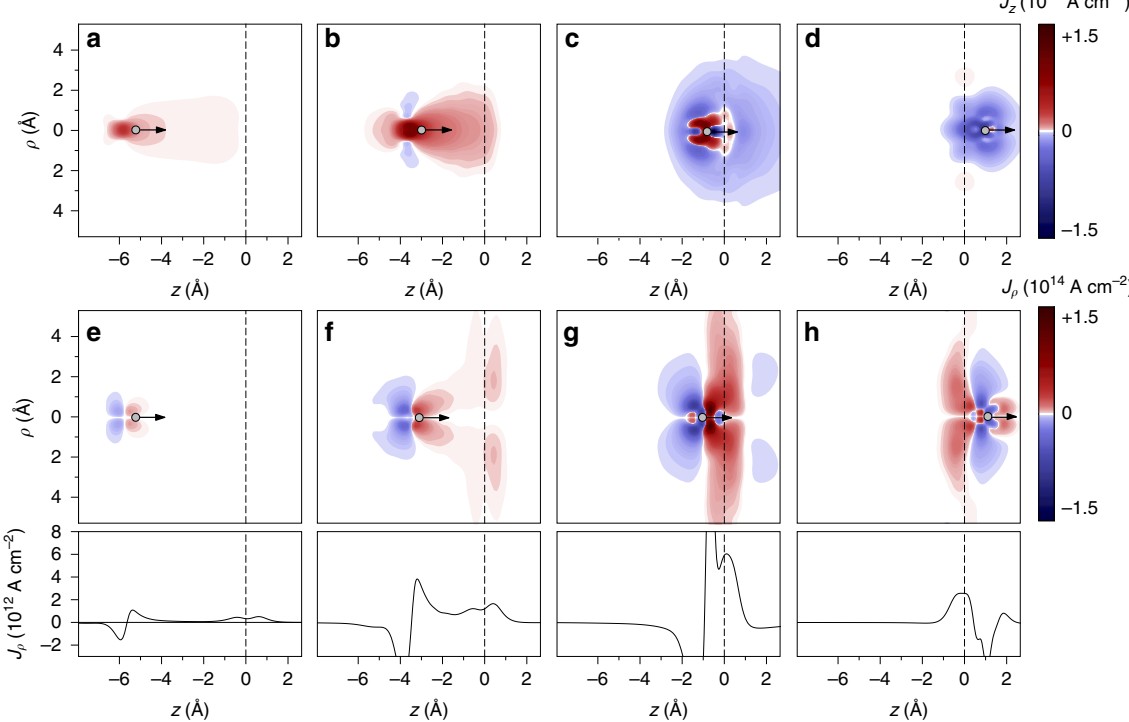

**Figure 2 | Perpendicular and radial current density obtained by TDDFT calculations.** Snapshots of the perpendicular $J_z$ (**a–d**) and radial $J_\rho$ (**e–h**) components of the current density for $q_{in} = 20$ at four different HCI–graphene distances obtained from TDDFT calculations performed in cylindrical $(\rho,z)$ coordinates with z-axis set along the projectile trajectory perpendicular to the target surface. The figures show that already above the graphene layer electrons are transferred to the approaching HCI and the current density along the direction of motion explains the charge exchange of the HCI. Extremely high transverse current density (**f–h**) along the graphene layer is obtained. The corresponding profiles (lower panels) show the z-dependent transverse current density averaged over a circle of 10 Å in radius. Values exceeding $10^{12}$ A cm$^{-2}$ are obtained. The position of the HCI is indicated by a small circle, the position of the graphene layer by the vertical dashed line.

presented for four different HCI–graphene distances $z_{ion}$ along the ingoing trajectory path. We use cylindrical $(\rho, z)$ coordinates, with the z axis set along the projectile trajectory assuming normal incidence geometry. Note that because of the cylindrical symmetry in our calculations (see Supplementary Note 2 for details), there is no azimuthal angular dependence. Already at $z_{ion} \sim -6$ Å charge transfer from graphene to the HCI is sizable. For $z_{ion} \sim -3$ Å, the $J_z$ component of the current density along the direction of motion is further increased and a significant transverse current density along the graphene layer (located at $z = 0$) can be observed.

The current density along the direction of motion $J_z$ determines the charge exchange and energy loss of the HCI (see the section Energy loss), while the current density along the transverse direction $J_\rho$ shows the fast local response of graphene to the strong HCI perturbation. In fact, our calculations reproduce our experimental estimate since the transverse current densities in the proximity of the graphene layer reaches values as high as $10^{12}$ A cm$^{-2}$. This means that, transiently and locally, in the femtosecond and nanometre scale, graphene is able to sustain extremely high current densities. The positive charges created by electron capture and electron emission are spread over the entire layer[44].

**Energy loss.** As neutralization is incomplete in our measurements charge state effects on the energy loss become experimentally accessible. As observed earlier with thicker foils[31,34,45], the energy loss of a HCI passing through thin sheets strongly depends on the number of electrons transferred to the ion. The energy loss as

experimentally deduced from the positions of the peaks for exit charge states $q_{out} = 2$ and $q_{out} = 4$ increases quadratically with the incident charge state (Fig. 3a). We observe keV energy losses, which are more than an order of magnitude larger than the result from a TRIM simulation (nuclear and electronic stopping) for neutral Xe transmitted through a layer with areal density of $3.82 \times 10^{15}$ at cm$^{-2}$ representing the target[46] (dashed line in Fig. 3a). This simulation predicts an energy loss of 228 eV taking into account our detector acceptance angle. The TRIM value is close to the energy loss expected from an extrapolation of our fit functions through our experimental data at $q_{in} = 0$ representing equilibrium stopping.

According to our TDDFT calculations, the non-adiabaticity of the charge-exchange and ionization processes introduced by the ion motion translates into an electronic energy loss of similar magnitude and charge state dependence as that observed in the measurements, although somewhat underestimated. The lower values in the theory as compared with the experiment are due to the fact that the used pseudo-potential description of the HCI does not include the full Coulomb singularity at distances $r$ smaller than a cutoff radius $R$. As discussed in the Supplementary Note 2 and shown in Supplementary Fig. 5, a simple change in the cutoff radius from $R = 0.53$ Å to $R = 0.26$ Å increases the value of the energy loss by a factor of two. Since the actual HCI would correspond to the value $R = 0$ and our aim is not to reproduce the data but to explain them, we consider our results rather satisfactory. In addition, the overestimation of the minimum excitation energy of graphene valence electrons in the jellium model is another reason for the underestimation of the energy loss, as it has been explicitly checked for helium

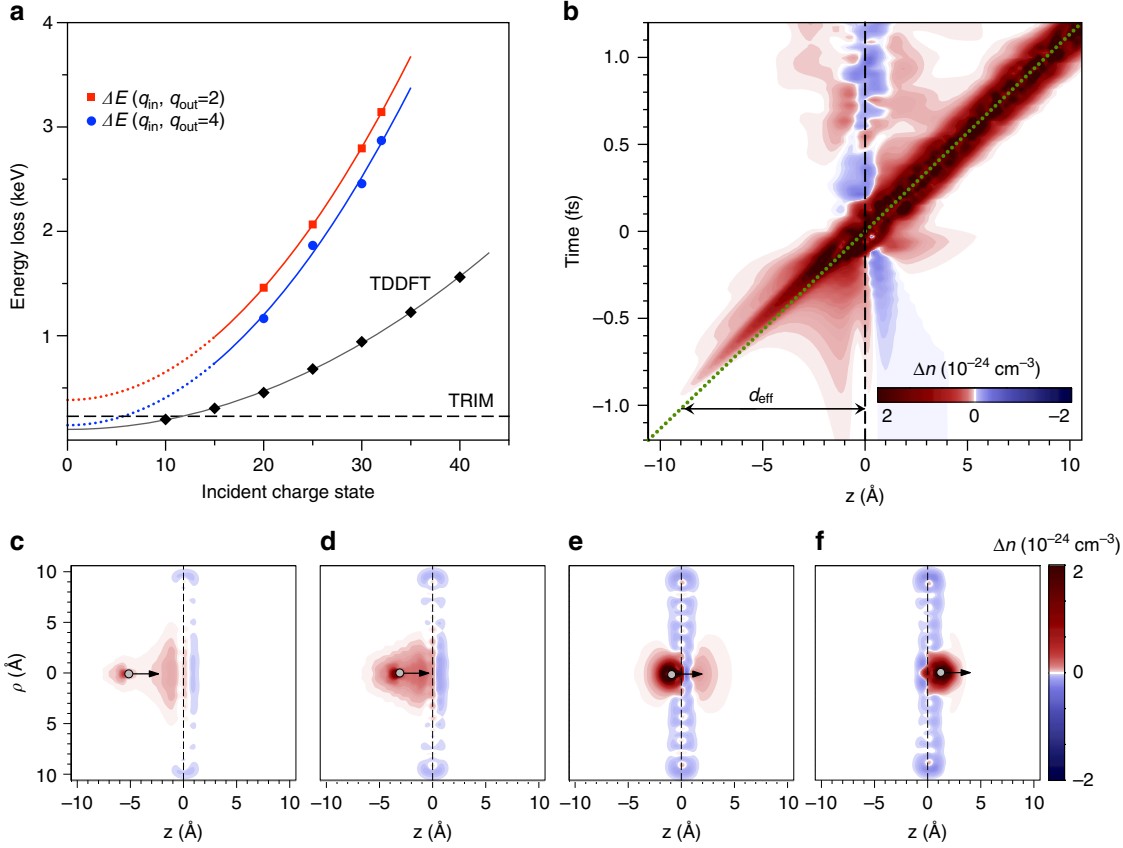

**Figure 3 | Ion energy loss results and a detailed view of the neutralization dynamics.** (**a**) Experimentally determined energy loss for ions with exit charge state $q_{out} = 2$ and $q_{out} = 4$ as a function of the incident charge state $q_{in}$. The energy of the projectiles was kept constant for all $q_{in}$ at $E = 40$ keV. The data points are fitted by a quadratic function and the dashed line shows the result from a TRIM simulation (nuclear and electronic stopping) for a graphite layer of 3 Å thickness. The experimental data are compared with results from TDDFT calculations that reproduce the parabolic dependence with the initial charge state and the order of magnitude of the energy loss. (**b**) The induced charge density along the $z$-axis perpendicular to the surface and passing through the ion centre as function of time for $q_{in} = 20$ at $v = 0.87$ nm fs$^{-1}$ using the TDDFT description. It shows that already $\sim 9$ Å above the graphene layer the HCI starts to capture electrons. It is also visible that the induced charge is not centred at the HCI position but lacks behind (see green dashed line). By approaching the surface more and more electrons are captured. (**c–f**) Snapshots of the induced charge density in cylindrical ($\rho,z$) coordinates for four different position of the incoming HCI projectile. (Supplementary Movie 1). The polarization of the surface due to the approaching HCI and the excitation of the graphene layer while and after the crossing of the ion are clearly visible as well as the HCI neutralization.

ions (Supplementary Note 2; Supplementary Figs 2 and 3). Reproducing the $q_{in}^2$ dependence and the order of magnitude of measured energy losses by our TDDFT calculations shows again that electronic response of the simulated system is well described.

The fact that the measured projectile energy loss can be assigned to the electronic excitations agrees with results obtained in ab initio studies for low projectile charges[38]. It provides a consistent link between charge transfer and energy loss processes and allows to explain the absence of the induced damage despite the large energy deposition. Indeed, owing to the high electron mobility of graphene the positive charges created in the local surface area by electron capture and electron emission into vacuum are promptly screened thus reducing the local electronic temperature.

A detailed view on the neutralization dynamics is given in Fig. 3b, where we show a two-dimensional plot of the induced density along the direction of the ion motion at different ion positions from a simulation done for $q_{in} = 20$ using our TDDFT description. The horizontal axis represents the distance to the graphene layer located at $z = 0$ and the vertical axis corresponds to the timescale. The HCI starts capturing electrons from graphene already at 9 Å. The strong attractive potential accelerates electrons towards the HCI and, approximately, half

of them end up captured by it along the incoming path before penetration into the graphene sheet. The induced electronic charge density as the HCI approaches the graphene layer has two components (Fig. 3c–f): one is formed by the convoy electrons around the HCI position, forming an asymmetric wake potential that slows down the ion, and the other one is located at the graphene layer due to the target polarization. Both components merge as the HCI gets closer to the target (Fig. 3e) and forward electron emission starts. Finally, after crossing the layer (Fig. 3f) the projectile is nearly neutralized and the corresponding induced electronic charge is centred around the HCI along its outgoing path. The actual HCI is probably not fully relaxed at the instant of crossing the graphene layer and, therefore, it still suffers a number of autoionization processes (not described in TDDFT) and subsequent de-excitation without a significant energy loss.

## Discussion

We have studied the electronic response of SLG to a large external field of an approaching HCI. We find an ultrafast neutralization within a few femtoseconds timescale leading to the capture and stabilization of almost (90%) all the missing electrons in the projectile. Our experiments and TDDFT calculations, both

suggest local current densities in the graphene plane exceeding $10^{12}$ A cm$^{-2}$, at least three orders of magnitude higher than previously established local breakdown currents, however, on a timescale of a few femtoseconds only. The exceptional electronic properties of graphene allow for a resupply of charge and distribution of the impact energy promptly enough to prevent Coulomb explosion in the electron-depleted region. In addition, the surprisingly large energy loss of the ion of a few keV, which is strongly connected with the charge-exchange process and depends on the incident and exit charge state, could be successfully explained. Our study revealed how graphene responds to extremely high fields and our results underline the exceptional properties of graphene for ultrafast electronic applications at high current densities.

## Methods

**Experimental set-up.** The measurements are performed at the Ion Beam Center of the Helmholtz-Zentrum Dresden-Rossendorf. Highly charged Xe ions are produced in a room temperature electron beam ion trap, charge state separated by an analysing magnet and then guided by several electrostatic lenses into the target chamber. Due to an electrostatic deceleration system, the kinetic energy of the extracted Xe ions can be varied between 0.1 and 4.4 $q$ keV corresponding to velocities between 0.13 and 0.5 nm fs$^{-1}$. The pressure in the experimental chamber is kept below $5 \times 10^{-9}$ mbar during measurements to prevent charge-exchange processes of the ions before interaction with the target.

The freestanding SLG sheets, which span over a regular array of holes in a TEM grid, are produced at the University Duisburg-Essen and transferred without the use of polymer coating (see Supplementary Note 1 and Supplementary Fig. 1 for details on the sample preparation and characterization). Before performing the transmission measurements, the graphene sheets are inspected by STEM to check the sample coverage and the grade of residual contamination. Contamination of the surface by water was either not present or does not affect the charge-exchange processes as the results of the experiments did not change when heating the samples up to 200 °C in ultra-high vacuum before and during the measurements. A heatable target holder and an electrostatic analyser are mounted in the target chamber (Fig. 1c). The electrostatic analyser is equipped with two channeltrons to analyse the charge state and energy of the transmitted ions and to count neutralized particles in forward direction. The analyser has an acceptance angle of 1.6°, the energy resolution was determined to be $\Delta E/E \approx 1.5 \times 10^{-3}$. The analyser's maximum operation voltage of 5,000 V limits the range of measurable charge states for projectiles with larger kinetic energies.

**TDDFT calculations.** Our TDDFT simulations are done following the time evolution of the Kohn–Sham orbitals (see Supplementary Note 2 for details) of the system defined by the constant velocity approach of a model pseudo-potential HCI (cutoff radius $R = 0.26$ Å, Coulomb tail $Q/r$) and a planar jellium disk with the correct work function value (4.6 eV)[13,47] representing the graphene layer. A non-uniform real space grid in cylindrical coordinates is used to treat properly the Coulomb singularity close to $r = R$. Finite size effects have been checked using jellium disks of different sizes containing 500, 1,004 and 2,000 electrons (Supplementary Figs 6 and 7). The Gunnarson and Lundqvist approximation[48] for the exchange correlation Kernel was used.

**Data availability.** The data that support the findings of this study are available from the corresponding author upon request.

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

## Acknowledgements

We acknowledge funding by Austrian Science Fund (FWF): project number: I1114-N20 and the German DFG (project number:WI 4691/1-1). We also acknowledge partial financial support from Gobierno Vasco project number IT-756-13 and MINECO project number FIS2013-48286-C2-1-P. We further acknowledge funding and fruitful discussions within the SPP 1495 'Graphene' and the collaborative research centre SFB 1242 'Non-equilibrium dynamics in condensed matter in the time domain' funded by the DFG. B.C.B. acknowledges funding from the European Union's Horizon 2020 research and innovation programme under the Marie Skłodowska-Curie grant agreement no. 656214-2DInterFOX. We are grateful for discussions with C. Lemell, J. Burgdörfer, P. Tiwald and I. Floss. We thank M. Heidelmann from the Interdisciplinary Center for Analytics on the Nanoscale (ICAN, core facility funded by the German Research Foundation, DFG) for support with the TEM Measurements.

## Author contributions

E.G., R.A.W. and V.S. performed the measurements, F.A. and S.F. were involved in planning and supervised the work, E.G. and R.A.W. processed the experimental data, performed the analysis, drafted the manuscript and designed the figures. R.P., I.A., A.K.K., A.A. and A.G.B. performed the TDDFT calculations. R.K., A.H. and M.S. manufactured the samples and characterized them with Raman spectroscopy and TEM, B.C.B. performed the STEM characterization. F.L. and A.V.K. aided in interpreting the results and worked on the manuscript. All authors discussed the results and commented on the manuscript.

## Additional information

**Competing financial interests:** The authors declare no competing financial interests.

**Publisher's note**: 

