## [Peer Review File · Nature Communications]

Reviewers' comments:

Reviewer #1 (Remarks to the Author):

The paper deals with femtosecond-timescale electronic response of graphene to a strong and localized electric field. The claims are that charge carriers in graphene can locally (\sim nm) move within few femtoseconds. Resulting in current densities as high as 10^{12} A/cm².

The experiment is performed in the following way. Free-standing graphene is bombarded by highly ($q \sim$ few tens) charged Xe ions (HCI) at \sim 100keV energy. Transmitted Xe ions are analyzed in respect to charge neutralization and energy loss. The ions speed is below 0.5 nm/fs. The experiment shows that transmitted HCIs are partially neutralized and decelerated. The charge neutralization is dependent on ion's speed and initial charge. The deceleration is observed to be larger (\sim 2keV vs \sim 0.2keV) than predicted by TRIM simulations for monocharged ions. In order to explain observed phenomena and to get more insights, TDDFT modeling and simulation is done. It shows that electrons are extracted from graphene when HCI is approaching closer than 0.5nm to graphene. Then, via Coulomb interaction HCI are further neutralized and decelerated. The TDDFT calculations are in a good agreement with mentioned above experimental observations. Thus, interaction time between HCIs and graphene membrane in reported experiments is \sim few fs. Over these few fs the HCI gets partially neutralized giving an estimate on how many electrons can be locally extracted from the graphene per unit time. Further, authors argue that local graphene charging has to be neutralized by neighboring charges within the same fs timescale, otherwise graphene lattice would explode and a nanopore would appear. A TEM image of bombarded graphene with no evidence of any defects created by HCIs is presented.

This referee finds it difficult to understand from the manuscript - why graphene has to neutralize the local charge on the same few-fs timescale as HCI is pathing through the graphene. Also the fact that TEM did not show any defects in bombarded graphene is surprising. Previous experiments and binary collision model has shown that for example monocharged Helium ions can penetrate 2D materials without creating damage with high probability, close to 99% depending on acceleration voltage (few tens keV) [1-3]. In its turn larger ions like Ga would create defects in freestanding graphene in more than 50% incidence cases. Xe-ions are even larger than Ga, thus it is expected that almost all Xe-ion interactions with graphene would create defects. Why these defects are not observed in the TEM characterization?

Authors argue that previously it was reported that HCI of Xe going through nm-thin carbon membranes would create nm-size pores. One can visualize this process by collision cascade effect resulting in transferring to each next membrane atom enough energy to be sputtered. This effect is present only in samples having finite volume. For 2D materials there is no collision cascade expected. Only binary collision events take place. Thus, it does not look to be necessary assuming "Coulomb explosions" effect in order to explain the difference between nm-pores in finite thickness membranes and only individual defects in 2D materials.

As a next step, authors estimate current density necessary to neutralize local charges in graphene and it turns out to be several orders of magnitude higher than previously reported. To avoid misunderstanding, it should be well explained in the manuscript that previously reported critical current was measured in DC mode and Joule heating is responsible for the break-down. In presented work, the current flows locally only for a few fs. Thus, there is no big surprise that the value for quick pulse current is much higher than the measured critical current under static conditions.

One of the interesting numbers in graphene is Fermi velocity, which is \sim 10^6 m/s. Presented work actually argues about moving charges in graphene on nm/fs speed which is very close to Fermi velocity. Could Authors comment on this. Is it possible to design an experiment where one would observe a threshold in behavior? If current for local charging neutralization in graphene will be higher than Fermi velocity, would then proposed by authors "Coulomb explosion" occur?

The reason of statistical variations in the HCIs neutralization amount is not clear from the paper. Fig 1(a) presents data where HCIs under the same conditions are neutralized by a different amount of charge. For further discussion average (or mean) number of neutralization charges is used. For example, when 60keV HCI with $q(\text{in})=30$ transmitted through graphene, its $q(\text{out})$

ranges between 2 and 14 with the peak value around 6. The distribution actually is not symmetric and quite broad. What can be a cause of this statistical variation? Can it come from contaminations on graphene surface?

To conclude, this referee finds the manuscript very interesting and it can be very valuable for the community and even the wider field after revisions suggested above.

Refs:

[1] J. Kotakoski, et al., "Towards two-dimensional all-carbon heterostructures via beam patterning of single-layer graphene" *Nano Lett* 15, 5944 (2015)

[2] J. Buchheim, et al., "Understanding the Interaction between Energetic Ions and Freestanding Graphene towards Practical 2D Perforation" *Nanoscale* 8, 8345 (2016)

[3] D. S. Fox, et al., "Nanopatterning and electrical tuning of MoS₂ layers with a subnanometer Helium ion beam" *Nano Letters*, 15, 5307 (2015)

Reviewer #2 (Remarks to the Author):

First I would like to congratulate the authors to a nice set of experiments with clear accompanying theory. I think that a broad audience in materials' science will be interested in the results since a clear example of how multiply charged ions are neutralised by a (quasi-)free standing graphene layer. Besides the experiments a simple theoretical model of the process is provided which will be a valuable reference for further work in the field.

Although the findings are discussed in view of the literature, I found a lack of further context as the problem of ion neutralisation is a topic of wide interest in the part of the graphene community that deals with physical vapour deposition onto graphene. The manuscript would benefit if this is mentioned, as interactions with low energy projectiles (from 1 to 100's of eV's) are discussed there — as a contrast to the 10-100 keV region discussed presently. References 15 to 19 covers this in part but more recent experimental work should be mentioned, e.g.:

P. Ahlberg, et al., *APL Mater.* 4., 046104 (2016)

G. Lupina, et al., *Appl. Phys. Lett.* 108 (2016)

Furthermore, the authors discuss their findings on carbon nano membranes (line 32), it should be appropriate to discuss/mention "mechanism of the defect formation in supported graphene by energetic heavy ion irradiation: the substrate effect", W. Li, et al., *Scientific Reports* 5, 9935 (2015).

Besides the points above, the manuscript is clearly written and presents experiment and theory in a compact yet thorough manner (especially with the extensive SI provided).

Response to referee comments on the manuscript NCOMMS-16-22142-T "Ultrafast electronic response of graphene to a strong and localized electric field" by E. Gruber *et al.*

We thank the reviewers for their positive reports and constructive comments on the manuscript. In our revised manuscript we have considered all comments and highlighted the changes made.

Response to Referee 1

1. This referee finds it difficult to understand from the manuscript - why graphene has to neutralize the local charge on the same few-fs timescale as HCl is pathing through the graphene.

From our experiments we can clearly conclude that:

- (i) the neutralization of the graphene sheet has to be fast enough to prevent Coulomb explosion (about 100 fs), since careful investigations of irradiated graphene samples did not show extended structural modifications
- (ii) the neutralization of the ion does not stop as the ion charge state increases, which reflects the high electron mobility in graphene, as well as the high current densities to both refill the nm spot around the impact point and bring additional electrons from nearby carbon atoms

Additionally, from our TDDFT calculations we see that the charge dissipates along the graphene sheet on a time scale similar to the ion passage time of some fs but this latter depends on the ion speed, while the former depends on intrinsic properties of graphene valence electrons, as discussed in both in the manuscript and, more deeply, in the supplementary material.

To make this more clear in the manuscript the sentences (a) on page 4 and (b) on page 5 are modified:

- (a) "*(ii) the charge neutrality of the interaction region is restored quickly after the ion passage..*" is replaced by

(ii) the electron flow along the graphene layer compensates the electron extraction by the HCl on the time scale of the collision (fs), otherwise the neutralization of the projectile would be stopped by the local charging of the target.

- (b) "*Indeed, owing to the high electron mobility of graphene the positive charges created in the local surface area by electron capture and electron emission into vacuum are promptly neutralized thus reducing the electronic temperature*".

Indeed, owing to the high electron mobility of graphene the positive charges created in the local surface area by electron capture and electron emission into vacuum are promptly screened thus reducing the electronic temperature.

- 2. Also the fact that TEM did not show any defects in bombarded graphene is surprising. Previous experiments and binary collision model has shown that for example monocharged Helium ions can penetrate 2D materials without creating damage with high probability, close to 99% depending on acceleration voltage (few tens keV) [1-3]. In its turn larger ions like Ga would create defects in freestanding graphene in more than 50% incidence cases. Xe-ions are even larger than Ga, thus it is expected that almost all Xe-ion interactions with graphene would create defects. Why these defects are not observed in the TEM characterization? Authors argue that previously it was reported that HCl of Xe going through nm-thin carbon membranes would create nm-size pores. One can visualize this process by**

collision cascade effect resulting in transferring to each next membrane atom enough energy to be sputtered. This effect is present only in samples having finite volume. For 2D materials there is no collision cascade expected. Only binary collision events take place. Thus, it does not look to be necessary assuming “Coulomb explosions” effect in order to explain the difference between nm-pores in finite thickness membranes and only individual defects in 2D materials.

[1] J. Kotakoski, et al., “Towards two-dimensional all-carbon heterostructures via beam patterning of single-layer graphene” *Nano Lett* **15**, 5944 (2015)

[2] J. Buchheim, et al., “Understanding the Interaction between Energetic Ions and Freestanding

Graphene towards Practical 2D Perforation” *Nanoscale* **8**, 8345 (2016)

[3] D. S. Fox, et al., “Nanopatterning and electrical tuning of MoS₂ layers with a subnanometer Helium ion beam” *Nano Letters*, **15**, 5307 (2015)

At the beginning we were also quite surprised to see no pore formation after HCl irradiation. Regarding defect production two mechanisms have to be considered:

- (a) the nuclear stopping (elastic collisions) may result in direct knock out of carbon atoms, but less than one carbon atom is sputtered on average by a 10-100keV Xe ion according to Lehtinen *et al.* [O.Lehtinen, PRB **81**, 153401 (2010)]. Even if point defects are produced, they will likely disappear due to dissociation of ubiquitous hydrocarbon molecules (see e.g. R. Zan, *Nano Lett* **12**, 3936 (2012)).
- (b) electronic stopping/potential energy deposition may cause Coulomb explosion tearing larger holes into the membrane. Since no extended structural modifications could be observed it can be concluded that ultrafast electronic response of graphene prevents Coulomb explosion. This is in contrast to insulators, like carbon nano membranes, wherein the neutralization of the membrane cannot be restored fast enough due the lower electron mobility which results in the formation of pores with diameters up to a few nm after exposure to HCl.
- (c) the mentioned references show defect formation of single layer graphene [1,2] or MoS₂ [3] by using HIM or FIB. In [2] it is shown that a total ion dose of Ga⁺ ions of $\sim 10^{15}$ cm⁻² is necessary to create a pattern in the monolayer of graphene.
We are investigating single ion impacts by using much lower fluences in the order of $\sim 10^{12}$ ions/cm² or less.

So far only the defect formation due to electronic stopping/potential energy deposition is mentioned in the manuscript. Considering the comment of the referee we now also mention possible defect formation by nuclear stopping by including the following sentence on page 2 in the manuscript:

In our case, elastic collisions (nuclear stopping) may cause direct knock out of carbon atoms, but less than one carbon atom is sputtered on average by a 10-100keV Xe ions according to Lehtinen *et al.* [O.Lehtinen, PRB **81**, 153401 (2010)]. Even if point defects are produced, they will likely disappear due to dissociation of ubiquitous hydrocarbon molecules [R. Zan, *Nano Lett* **12**, 3936 (2012)].

We agree with the referee that it is appropriate to mention also the work of Kotakoski *et al.*, Buchheim *et al.* and Fox *et al.* which we have now done in the introduction (page 2).

3. As a next step, authors estimate current density necessary to neutralize local charges in graphene and it turns out to be several orders of magnitude higher than previously reported. To avoid misunderstanding, it should be well explained in the manuscript that previously reported critical current was measured in DC mode and Joule heating is

responsible for the break-down. In presented work, the current flows locally only for a few fs. Thus, there is no big surprise that the value for quick pulse current is much higher than the measured critical current under static conditions.

We agree with the referee. To avoid a misunderstanding we have modified the text on two occasions:

(a) The sentence in the introduction (page 1) now reads:

DC measurements on supported single layer graphene (SLG) reveal breakdown current due to Joule heating larger than in copper, with densities of about 10^8 - 10^9 Acm⁻².

(b) Moreover, we mention in the summary (page 8) that the high current value, which we estimate from our experimental data and which we obtain by the TDDFT calculations, is limited to a fs time scale:

Our experiments and TDDFT calculations, both suggest local current densities in the graphene plane exceeding 10^{12} Acm⁻², at least three orders of magnitude higher than previously established local breakdown currents, however on a time scale of a few fs only.

- 4. One of the interesting numbers in graphene is Fermi velocity, which is $\sim 10^6$ m/s. Presented work actually argues about moving charges in graphene on nm/fs speed which is very close to Fermi velocity. Could Authors comment on this. Is it possible to design an experiment where one would observe a threshold in behavior? If current for local charging neutralization in graphene will be higher than Fermi velocity, would then proposed by authors "Coulomb explosion" occur?**

We actually thought about such a possibility. However the impact of a Xe ion in charge state 20+ creates a charge depletion region with the diameter of typically 1nm only on the graphene layer (see fig 3 (c-f) in the manuscript). In this case a charge (electron) speed of nm/fs is high enough to restore the neutralization of the graphene layer before Coulomb explosion sets in. To observe Coulomb explosion the formation of much larger charge depletion areas (in the range of 100nm in diameter) would be necessary ($v_{\text{Fermi}} \sim 1$ nm/fs; $t_{\text{CoulExp}} = 100$ fs $\rightarrow r = 100$ nm). It seems unlikely to achieve such large excited areas with highly charged ions, even by using ions in much higher charge states (of course this has to be verified experimentally, but this was out of the scope of the present paper). A better choice, however, would be the use of high power lasers, which are able to ionize an area of several hundred to thousands of nm² with ultrashort pulses.

- 5. The reason of statistical variations in the HCIs neutralization amount is not clear from the paper. Fig 1(a) presents data where HCIs under the same conditions are neutralized by a different amount of charge. For further discussion average (or mean) number of neutralization charges is used. For example, when 60keV HCl with $q(\text{in})=30$ transmitted through graphene, its $q(\text{out})$ ranges between 2 and 14 with the peak value around 6. The distribution actually is not symmetric and quite broad. What can be a cause of this statistical variation? Can it come from contaminations on graphene surface?**

Fig 1 a in the manuscript shows typical transmission spectra. The measured spectra are broadened by the design of the electrostatic analyzer. To extract the correct abundances and widths, every single peak is fitted by a convolution of a rectangular function representing the entrance slit and a Moyal

function (details on the data evaluation can be found in the supplementary data of [R. Wilhelm, *2D Mater.* **2** 035009 (2015)]). The figure below shows the corrected abundances as a function of the exit charge state for a fixed kinetic energy of $E_{kin}=40\text{keV}$ for various incident charge states. It can be seen that the charge distributions follow a symmetric Gaussian function with a FWHM of $\sim 3\text{-}5$ electrons. From these Gaussian functions the mean exit charge state is evaluated.

After the passage of the graphene layer, the ion has captured many electrons (close to neutral) but can still be in a somewhat excited state. The final relaxation into the ground state can be a post-collision process, where non-radiative (i.e. electron emitting) and radiative de-excitation processes (without electron emission) will compete. This will lead to slightly different exit charge states as measured in the asymptotic region.

Contaminations on the graphene are comparable with thicker foils like the 1nm thick carbon nano membranes investigated in [R. Wilhelm, *Phys.Rev.Lett* **112**,153201 (2014)]. Contaminations therefore would result on average in neutral or singly charged projectiles.

Figure 1: Normalized and corrected abundance of $Xe^{q_{out}+}$ exit charge states for different incident charge states at a constant kinetic energy $E_{kin}=40\text{keV}$. The solid lines are Gaussian fits through the data points.

In the manuscript the following sentence is added as a footnote (page 4):

To extract the abundances and widths of every single peak, the spectra have to be deconvoluted with the analyzer function, since the spectra are broadened by the design of the electrostatic analyzer (details on the data evaluation can be found in the supplementary data in [R. Wilhelm, *2D Mater.* **2** 035009 (2015),]). The corrected abundances follow a symmetric Gaussian function with a FWHM of $\sim 3\text{-}5$ electrons as a result of final de-excitation processes. From these Gaussian fits the mean value q_{out} is extracted.

Response to Referee 2:

Although the findings are discussed in view of the literature, I found a lack of further context as the problem of ion neutralisation is a topic of wide interest in the part of the graphene community that deals with physical vapour deposition onto graphene. The manuscript would benefit if this is mentioned, as interactions with low energy projectiles (from 1 to 100's of eV's) are discussed there — as a contrast to the 10-100 keV region discussed presently. References 15 to 19 covers this in part but more recent experimental work should be mentioned, e.g.:
P. Ahlberg, et al., APL Mater. 4., 046104 (2016)
G. Lupina, et al., Appl. Phys. Lett. 108 (2016)

Furthermore, the authors discuss their findings on carbon nano membranes (line 32), it should be appropriate to discuss/mention “mechanism of the defect formation in supported graphene by energetic heavy ion irradiation: the substrate effect”, W. Li, et al., Scientific Reports 5, 9935 (2015).

We thank the referee for the favorable report and agree that it is appropriate to mention the work of Ahlberg *et al.* and Lupina *et al.* which we have now done in the introduction (page 2).

We agree with the referee that for the sake of completeness also the work of Li *et al.* should be mentioned. We included on page 2 the following sentence:

Note that for the free-standing graphene, in difference to supported graphene layers, the defect formation because of a collision cascade in the substrate [W. Li, et al., Scientific Reports 5, 9935 (2015)] is not operative.

REVIEWERS' COMMENTS:

Reviewer #1 (Remarks to the Author):

This referee would like to thank the Authors for the answers and clarifications. The manuscript in its current state is recommended for publication in Nature Communications.

Minor (optional) remark:

Comparing Fermi velocity in graphene to nm/fs speeds extracted from the presented experiment in principle leads to a conclusion that "Ultrafast" in the title can be extended to "ultimately fast".

Minor (optional) remark:

Comparing Fermi velocity in graphene to nm/fs speeds extracted from the presented experiment in principle leads to a conclusion that “Ultrafast” in the title can be extended to “ultimately fast”.

We thank the referee for the favorable report. In principle we agree with the referee, but nevertheless we decided to keep the title as it is, because we do not want to further stress this point in the paper.